

# Metabolic shift and the effect of mitochondrial respiration on the osteogenic differentiation of dental pulp stem cells

Lingyun Wan[1,*], Linyan Wang[2,*], Ran Cheng[1], Li Cheng[1] and Tao Hu[1]

[1] State Key Laboratory of Oral Diseases, Frontier Innovation Center for Dental Medicine Plus, National Clinical Research Center for Oral Diseases, West China Hospital of Stomatology, Sichuan University, Chengdu, Sichuan, China

[2] Chengdu Second People's Hospital, Chengdu, Sichuan, China

[*] These authors contributed equally to this work.

Corresponding authors
Li Cheng, dentistcl@scu.edu.cn
Tao Hu, hutao@scu.edu.cn

## ABSTRACT

**Background**. Metabolism shifts from glycolysis to mitochondrial oxidative phosphorylation are vital during the differentiation of stem cells. Mitochondria have a direct function in differentiation. However, the metabolic shift and the effect of mitochondria in regulating the osteogenic differentiation of human dental pulp stem cells (hDPSCs) remain unclear.

**Methods**. Human dental pulp stem cells were collected from five healthy donors. Osteogenic differentiation was induced by osteogenic induction medium. The activities of alkaline phosphatase, hexokinase, pyruvate kinase, and lactate dehydrogenase were analyzed by enzymatic activity kits. The extracellular acidification rate and the mitochondrial oxygen consumption rate were measured. The mRNA levels of *COL-1, ALP, TFAM,* and *NRF1* were analyzed. The protein levels of p-AMPK and AMPK were detected by western blotting.

**Results**. Glycolysis decreased after a slight increase, while mitochondrial oxidative phosphorylation continued to increase when cells grew in osteogenic induction medium. Therefore, the metabolism of differentiating cells switched to mitochondrial respiration. Next, inhibiting mitochondrial respiration with carbonyl cyanide-chlorophenylhydrazone, a mitochondrial uncoupler inhibited hDPSCs differentiation with less ALP activity and decreased *ALP* and *COL-1* mRNA expression. Furthermore, mitochondrial uncoupling led to AMPK activation. 5-Aminoimidazole-4-carboxamide ribonucleotide, an AMPK activator, simulated the effect of mitochondrial uncoupling by inhibiting osteogenic differentiation, mitochondrial biogenesis, and mitochondrial morphology. Mitochondrial uncoupling and activation of AMPK depressed mitochondrial oxidative phosphorylation and inhibited differentiation, suggesting that they may serve as regulators to halt osteogenic differentiation from impaired mitochondrial oxidative phosphorylation.

## INTRODUCTION

Human dental pulp stem cells (hDPSCs), a source of adult multipotent stem cells, can differentiate into multiple cell types, including odontoblasts/osteoblasts, chondrocytes, adipocytes, myogenic and neural cells (*Sui et al., 2020*; *Tsutsui, 2020*). Due to their easy isolation and great potential in tissue engineering and regenerative medicine, DPSCs are widely used in various fields (*Fernandes et al., 2020*). Numerous studies have been dedicated to uncovering the detailed mechanisms involved in their self-renewal ability and multilineage differentiation potential. The capacity of dental pulp stem cells to differentiate is essential for dental pulp repair and dentin rebuilding (*Sui et al., 2020*).

Energy metabolism plays a critical role in regulating the proliferation, differentiation, and many biological processes of stem cells. The relationship between biological activities and mitochondrial aerobic oxidative phosphorylation has been studied since the first study dated 1957 (*Green, Lester & Ziegler, 1957*). In stem cells, mitochondria produce energy not only to maintain homeostasis but also for differentiation (*Tsutsui, 2020*). The metabolism shifts between mitochondrial oxidative phosphorylation (OXPHOS) and glycolysis change along with the cell status and levels of mitochondrial maturation (*Khacho & Slack, 2018*; *Wanet et al., 2015*). For instance, glycolysis is predominant in undifferentiated stem cells, while differentiated cells rely more on OXPHOS (*Ly, Lynch & Ryall, 2020*). In mesenchymal stem cells, the mitochondrial process of OXPHOS is activated during osteogenic differentiation. However, the levels of glycolysis are maintained similarly to those in undifferentiated cells (*Shum et al., 2016*). Our previous study demonstrated that at the initial stage of hDPSCs differentiation, OXPHOS and glycolysis were upregulated (*Wang et al., 2016*). It should be determined whether mitochondrial and glycolytic changes exist during hDPSCs' continuous differentiation.

Moreover, the dynamic distribution, subcellular content, and structure of mitochondria have been shown to exhibit peculiar functions in the processes of cellular differentiation and reprogramming (*Khacho & Slack, 2017*). Mitochondrial-mediated shifts accompanied by mitochondrial remodeling and dynamics are vital to neural stem cell differentiation and fate (*Coelho et al., 2022*). Other questions that need to be addressed are the role of mitochondria in the differentiation of hDPSCs.

Some studies are underway to unveil the bidirectional crosstalk between mitochondria and cell differentiation, such as reactive oxygen species production, energy-sensing pathways, and the hypoxia-inducible factor pathway. Adenosine monophosphate (AMP)–activated protein kinase (AMPK), as a central metabolic sensor sensitive to the AMP/ATP ratio, can be activated by various types of cellular stress (*Aslam & Ladilov, 2022*; *Li et al., 2013*). AMPK regulates the fate of stem cells and serves as a critical regulator of differentiation (*Liu et al., 2021*; *Sun et al., 2017*; *Yang et al., 2016*). Previous reports have shown that AMPK is required for immune cell differentiation and adipogenic differentiation (*Rambold & Pearce, 2018*; *Son et al., 2019*; *Yang et al., 2016*). AMPK influenced the osteogenic differentiation of human dental pulp mesenchymal stem cells (*Pantovic et al., 2013*). The study also detected the change in AMPK when OXPHOS was intervened.

Metabolism is a vital regulator of stem cells and their fate. Anaerobic glycolysis may be an adaptation to the low oxygen niche for stem cells and maintain their stemness. The shift from glycolysis to OXPHOS is required for stem cells to differentiate. The reverse transition from OXPHOS to glycolysis is required to induce pluripotency from somatic cells (*Zhang, Menzies & Auwerx, 2018*). Regulating the metabolism of stem cells could be a potential target for tissue engineering or regeneration medicine.

This study aimed to investigate the changes in mitochondrial OXPHOS and glycolysis during hDPSCs osteogenic differentiation and their possible regulatory factors.

## MATERIALS & METHODS

### Reagents and antibodies

Hexokinase (HK), pyruvate kinase (PK), lactate dehydrogenase (LDH), and alkaline phosphatase (ALP) enzyme activity kits were obtained from Jiancheng (Nanjing, China). MitoTracker Red CMXRos (M7512) and Alexa Fluor 488 Phalloidin (A12379) were obtained from Invitrogen (Carlsbad, CA, USA). The XF Cell Energy Phenotype Test Kit, XFp extracellular flux analyzer, XFp culture microplates, and bicarbonate-free DMEM were obtained from Seahorse Bioscience Agilent Technologies (North Billerica, MA, USA). Hoechst 33342 (B2261) and carbonyl cyanide 3-chlorophenylhydrazone (CCCP, C2759) were purchased from Sigma–Aldrich (St Louis, MO, USA). The following antibodies were used: CD90 (FITC, mouse anti-human, BioLegend, 328107), CD105 (PerCP/Cy5.5, mouse anti-human, BioLegend, 323215), CD19 (PE, mouse anti-human, BioLegend, 982402), CD34 (PE, mouse anti-human, BioLegend, 343505), AMPK (Abcam, ab80039), p-AMPK (Abcam, ab133448), anti-GAPDH (Abcam, ab9485,ab8245), goat anti-rabbit IgG (H + L)-HRP (Bio-Rad Laboratories, 1706515), goat anti-mouse IgG (H + L)-HRP (Bio-Rad Laboratories,1706516) and goat anti-rabbit IgG (H+L) Alexa Fluor 594-conjugated secondary antibody (Invitrogen, R37117).

### Cell culture and treatment

After receiving written informed consent from patients and their parents, human dental pulp tissues were collected from caries- and periodontitis-free third molars extracted for orthodontic purposes from healthy donors ($n = 5$). The experimental procedures were conducted according to the Declaration of Helsinki. The protocol was approved by the Institutional Ethics Committee of West China Hospital of Stomatology (WCHSIRB-D-2017-052; WCHSIRB-D-2020-075-R1). After washing with sterile PBS, dental pulp tissues were cut into fragments, digested with 3 mg/mL type I collagenase at 37 °C for 30 min, and placed on 25 cm$^2$ culture dishes. Later, they were maintained in standard medium, which was low glucose Dulbecco's modified Eagle's medium (DMEM) containing 10% fetal bovine serum (FBS) (Gibco, Carlsbad, CA, USA), penicillin (100 units/mL), and streptomycin (100 mg/mL) at 37 °C in 5% $CO_2$. Cells were sub-cultured when they reached 80% confluence and used at passages 3∼6 (*Hu et al., 2022*). Osteogenic differentiation was induced by osteogenic induction medium (OIM), the standard medium supplemented with 10 mM b-glycerophosphate, 50 mg/mL ascorbic acid, and 100 nM dexamethasone

(Sigma–Aldrich, St. Louis, MO, USA) for 1, 3, 5, and 7 days (*Chen et al., 2013*). All media were replaced every two days.

To inhibit OXPHOS activity, cells were stimulated with a mitochondrial uncoupler, carbonyl cyanide 3-chlorophenylhydrazone (CCCP, 2 μM). A final concentration of 2 μM was used for this study according to a previous lethality analysis (LD50) (*Mandal et al., 2011*). To activate AMPK, cells were treated with a purine nucleoside analog, 5-aminoimidazole-4-carboxamide ribonucleotide (AICAR, 500 μM). When cells achieved 50%~60% confluency, AICAR was added to the cell medium at a concentration of 500 μM for 1, 3, 5, and 7 days (*Lee et al., 2010*).

## Flow cytometry analysis

Flow cytometry analysis was used to identify stem cell surface markers (*Aydin & Şahin, 2019*; *Ponnaiyan, 2014*). The primary cells at the third passage were detached from flasks using trypsin and washed twice with PBS. A total of $1\times10^6$ cells in 100 μL of PBS were labeled with 5 μL of antibodies against the surface markers CD90 (1:20), CD105 (1:20), CD19 (1:20), and CD34 (1:20) for 1 h at 4 °C protected from light. One sample without additional antibodies was used as a negative control. After the cells were washed twice and resuspended in 500 μL PBS, the labeled cells were analyzed using a Cytomics FC 500 flow cytometer (Beckman Coulter, Brea, CA, USA) and FlowJo software (Tree Star, Ashland, OR, USA).

## Alizarin red staining

For osteogenic differentiation, $1\times10^5$ hDPSCs per well were seeded in 6-well plates. At 60%~70% confluence, cells were cultured with OIM for 21 days and then washed with PBS. The cells were then stained with 40 mmol/L alizarin red S staining solution (pH 4.2, Sigma-Aldrich, St Louis, MO, A5533) for 10 min after being fixed with 4% paraformaldehyde for 20 min at room temperature (*Yin et al., 2021*).

## Enzymatic activity assay

For the ALP, PK, and LDH activity assays, $1\times10^6$ cells were resuspended in 0.5 mL of ice-cold PBS and ultrasonically decomposed 5 times at 300 W for 5 s/time with 30s intervals on ice. For the HK activity assay, $5\times10^6$ cells were resuspended in one mL of extracting solution and ultrasonically decomposed. Then, the supernatant was collected at 8,000 g/min for 10 min at 4 °C. The supernatant was used for protein quantification by the BCA protein assay (KGP902; keyGEN, Nanjing, China) and subsequent activity assays according to the manufacturer's instructions. The optical density values were determined at 450 wavelengths by a Varioskan Flash multi-plate reader (Thermo Scientific, Waltham, MA, USA). The mean values of the mean absorbance rates from three wells were calculated. Enzyme activities were calculated from optical density values and normalized by the total protein of each sample.

## Cell energy phenotype analysis

Our previous study showed that hDPSCs initiated differentiation on Day 3 (*Wang et al., 2016*). With further differentiation, extracellular flux rates were measured in this study.

Cells ($4\times10^3$ cells/well) were plated on XFp cell culture microplates and divided into experimental and control groups. The experimental group was cultured in OIM, while the control group was grown in standard medium at 37 °C and 5% $CO_2$. On Days 1, 3, 5, and 7, the extracellular acidification rate (ECAR, as an indicator of the glycolysis potential) and the mitochondrial oxygen consumption rate (OCR, as a parameter of mitochondrial respiration) were analyzed using the Seahorse XF Cell Energy Phenotype Test Kit on the Seahorse XFp Bioanalyzer. One hour before beginning the measurement, cells were switched to Seahorse bicarbonate-free DMEM, adjusted to pH 7.4 with NaOH, and subsequently incubated in a non-$CO_2$ incubator at 37 °C for 1 h. Then, OCR and ECAR tests were performed under baseline and stressed conditions after oligomycin (1 µM, an ATP synthase inhibitor) and carbonylcyanide-p-trifluoro-methoxyphenyl hydrazone (FCCP, 2 µM, a mitochondrial uncoupling agent) were added to the microplates. Extracellular flux rates were analyzed by using Seahorse XF software after normalization to the total protein units, and the ratio of OCR/ECAR was calculated.

### Real-time quantitative PCR (qPCR)

Total RNA was extracted from hDPSCs (4∼5 passages) using TRIzol (Invitrogen). RNA quantification and quality control were done using a 2000c Nanodrop spectrophotometer (Thermo Scientific, Waltham, MA, USA). cDNA was reverse transcribed with the PrimeScript RT Reagent Kit (Takara, Osaka, Japan). A negative control without reverse transcriptase was also implemented. Real-time PCR was performed using a SYBR Green PCR kit (Takara) and a Roche LightCycler480 Real-Time PCR System (Roche, Basel, Switzerland), as described previously (*Takeda et al., 2008*). Average crossing threshold (CT) values were calculated from the triplicate cDNA samples. Relative gene expression was calculated as $2^{-\triangle\triangle CT}$ (*Livak & Schmittgen, 2001*). GAPDH was used as the internal reference gene, and levels were relativized to the control group. The sequences of specific primers are shown in Table S1.

### Fluorescence microscopy

Cells were seeded on coverslips at $1\times10^4$ cells/well in 6-well plates and incubated with MitoTracker Red CMXRos (MRC) in FBS-free DMEM at a concentration of 100 nM at 37 °C for 30 min. After washing, cells were fixed in 4% paraformaldehyde for 10 min and then permeabilized with 0.05% Triton X-100 in PBS for 10 min at room temperature. Then, the cells were incubated with Alexa Fluor 488 phalloidin (1:50) to visualize the cytoskeleton for 1 h at room temperature. Nuclei were stained with Hoechst 33342 (200 µg/mL) for 5 min.

For the immunofluorescence experiment, cells were first fixed in 4% paraformaldehyde for 10 min and were permeabilized with 0.05% Triton X-100 in PBS for 10 min. After washing, cells were blocked with 5% BSA in PBS for 30 min and incubated with primary antibody against p-AMPK (*Khorraminejad-Shirazi et al., 2020*) (1:200) overnight at 4 °C. Following further washing, the cells were incubated with Alexa Fluor 594-conjugated secondary antibody (1:1200). Nuclei were also visualized by Hoechst 33342 (200 µg/mL) for 5 min. Fluorescence signals were obtained using an Olympus microscope (Olympus IX73; Olympus, Munster, Germany).

## Immunoblotting

According to the manufacturer's recommendation, total proteins from hDPSCs (5~6 passages) were extracted using a total protein extraction kit (KeyGEN BioTECh, Nanjing, China). SDS/PAGE was performed on Bio-Tris 5%–10% gradient polyacrylamide gels. Proteins were transferred to PVDF membranes (Bio-Rad Laboratories, Hercules, CA, USA), and membranes were blocked with 5% BSA or 3% nonfat milk in Tris-buffered saline containing 0.1% Tween (TBS-T) for 60 min at room temperature, according to the manufacturer's instructions. The membranes were incubated overnight with the appropriate primary antibodies, AMPK (1:1000), p-AMPK (1:1000; Abcam, Cambridge, UK), and anti-GAPDH (1:1000). Detection was subsequently performed with HRP-conjugated secondary antibodies for 1 h at room temperature, goat anti-rabbit IgG-HRP (1:10,000) and goat anti-mouse IgG-HRP (1:10,000). The membranes were scanned using the GelDoc XR+ System (Bio-Rad Laboratories, Hercules, CA, USA). The density of Western blot band signals was monitored using Quantity One software.

## Statistical analysis

Every experiment was independently replicated a minimum of three times. ANOVA and Student's t test were performed to determine statistical significance between the control and experimental groups using SPSS version 17.0 software. *P value* s were considered to be significant when $p < 0.05$ (*$p < 0.05$, ** $p < 0.01$, *** $p < 0.001$).

## RESULTS

### During osteogenic differentiation, glycolysis function declined after an initial slight increase

For the identification of hDPSCs, flow cytometry was used to analyze cell surface marker expression, and the cells were also incubated in OIM for alizarin red staining. The results showed that the mesenchymal stem cell markers CD90 and CD105 were 99.1% positive and 78.9% positive in hDPSCs, respectively. The hematopoietic stem cell markers CD19 and CD34 were merely 2.1% positive and 4.6% positive in hDPSCs. Alizarin red staining of the cells incubated in OIM was positive (Fig. S1).

To assess osteogenic differentiation, the mRNA expression of osteogenic differentiation markers (*ALP* and *COL-1*) and ALP activity were detected on Days 0, 1, 3, 5, and 7. By q-PCR, the expression of *ALP* mRNA increased at 3 days after differentiation induction ($p < 0.01$), and *COL-1* started to be upregulated 5 days after treatment ($p < 0.05$) compared with the control (Figs. 1A and 1B). ALP activity, a marker of osteogenic differentiation, was also monitored (*Qian et al., 2015*). The increase in ALP activity occurred from Day 3 and continued to increase during the process of differentiation (1.98 ± 0.1 *vs.* 5.63 ± 0.07; 2.86 ± 0.08 *vs.* 7.74 ± 0.07; 2.29 ± 0.11 *vs.* 11.25 ± 0.18, $p < 0.001$, Fig. 1C).

The activity of several enzymes in glycolysis was measured. The activities of HK ($p < 0.05$), PK ($p < 0.05$), and LDH ($p < 0.01$) slightly increased at Day 3 and then decreased to a significantly lower level than that of the control at Day 7 during the differentiation of hDPSCs (Figs. 1D, 1E, and 1F).

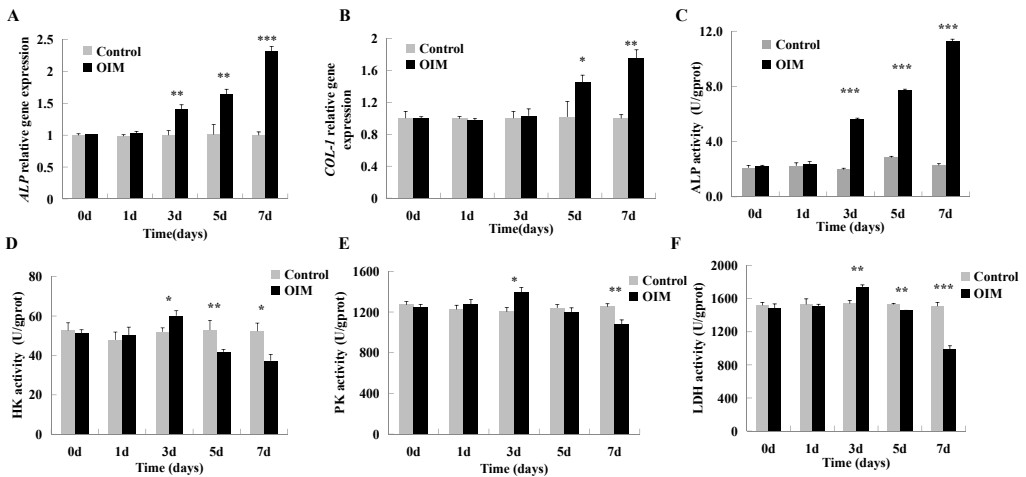

**Figure 1** **Glycolysis increased first and then decreased during hDPSCs differentiation.** Cells were cultured in control and OIM for 0, 1, 3, 5, and 7 days, respectively. (A, C) mRNA expression and activity of ALP were upregulated from day 3. (B) COL-1 was upregulated from day 5. (D) HK, a key glycolytic enzyme, increased at day 3 when cells were initiated to differentiate and decreased from day 5. (E, F) The same trend was detected in PK and LDH. Data were mean $\pm$ SD, * $p < 0.05$, ** $p < 0.01$, *** $p < 0.001$.

## Mitochondrial OXPHOS predominated in the energy production of the differentiating hDPSCs

The data showed that mitochondrial transcription Factor A (*TFAM*, controlling mtDNA expression and mitochondrial biogenesis) expression was elevated at Days 3, 5, and 7 compared to the control (Fig. 2A, $p < 0.05$, $p < 0.01$, $p < 0.01$). The expression of *TFAM* in the OIM was also gradually upregulated beginning on Day 3 (Fig. 2A, $p < 0.001$). However, the expression level of nuclear respiratory Factor 1 (*NRF1*, controlling nuclear-encoded respiratory chain component expression) was unchanged compared to that in control cells (Fig. 2B).

Later, the mitochondrial OCR (mitochondrial respiration indicator) and ECAR (glycolysis indicator) under baseline and stress conditions were analyzed by a Seahorse XF Cell Energy Phenotype Test Kit. Two stressor compounds, oligomycin, which causes an increase in glycolysis, and FCCP, which drives OCR rates higher, were used to induce stress conditions. Under basal conditions, consistent with upregulated *TFAM* mRNA expression, the OCR of differentiating cells was greater than that of the control on Day 3 (Fig. 2C). Nevertheless, the ratio of OCR/ECAR was not different (Fig. 2D). At Days 5 and 7, ECAR was significantly downregulated, and OCR was significantly increased. Moreover, the cells in OIM exhibited higher OCR/ECAR compared to the control at Days 5 and 7 (Fig. 2D, Day 5, $1.03 \pm 0.08$ *vs.* $2.71 \pm 0.39$, $p < 0.01$; Day 7, $1.06 \pm 0.17$ *vs.* $2.29 \pm 0.09$, $p < 0.01$). Under stressed metabolic conditions, a similar pattern of specific dynamic changes in mitochondrial OXPHOS and glycolysis was observed (Fig. 2E).
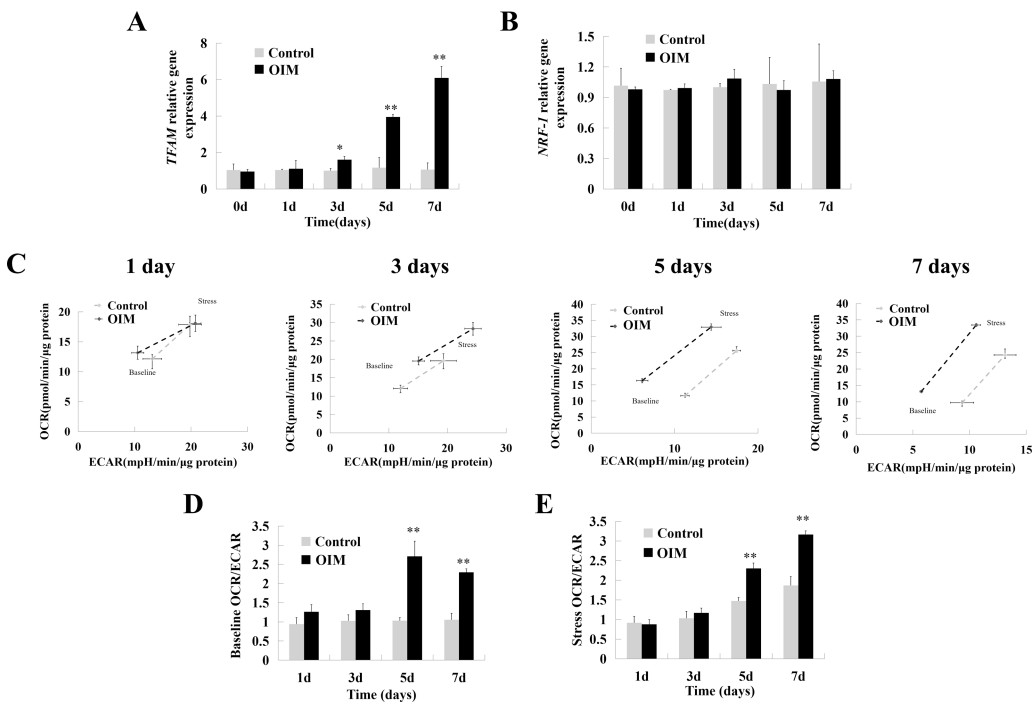

**Figure 2** **Cell energy phenotype shifted to mitochondrial OXPHOS as hDPSCs differentiated.** (A) *TFAM* was upregulated when hDPSCs differentiated. (B) *NRF1* was not changed upon differentiation. (C) On days 1, 5, and 7, the OCR and ECAR of cells growing in basal condition were first measured. Then those of stressed condition without Oligomycin and FCCP were assayed to show the max metabolic potential. The ECAR was on the horizontal axis, and the OCR was on the vertical axis. The hollow rhombus was for basal condition, while the solid was for stressed condition. (D) For the differentiating cells, the ratio of basal OCR/ECAR, which was calculated to show the major energy phenotype, increased from day 5. (E) The stressed OCR/ECAR ratio had a similar tendency. Data were mean ± SD, * $p < 0.05$, ** $p < 0.01$, *** $p < 0.001$.

## Mitochondrial uncoupling interfered with hDPSCs' osteogenic differentiation

A mitochondrial uncoupler, CCCP, was used to attenuate mitochondrial respiration (*Kane et al., 2018*). The cells were cultured in OIM with or without CCCP. Upon differentiation, both the basal and stressed mitochondrial OCRs were depressed after CCCP treatment compared to cells cultured in OIM (Fig. 3A). The ECAR of cells growing in the presence of CCCP showed a decrease compared to the control at Day 3 in the baseline condition ($21.21 \pm 0.9$ *vs.* $13.71 \pm 1.25$, $p < 0.01$). At the same time, there was no difference on other days (Fig. 3B). With MitoTracker Red CMXRos staining, the mitochondrial morphology was punctate. It failed to form a functional network (yellow arrow, Fig. 3C). Importantly, this suppression of mitochondrial respiration attenuated the differentiation of hDPSCs, manifested as inhibited ALP activity and restrained *ALP* and *COL-1* mRNA expression, compared to cells growing in the absence of CCCP (Fig. 3D).

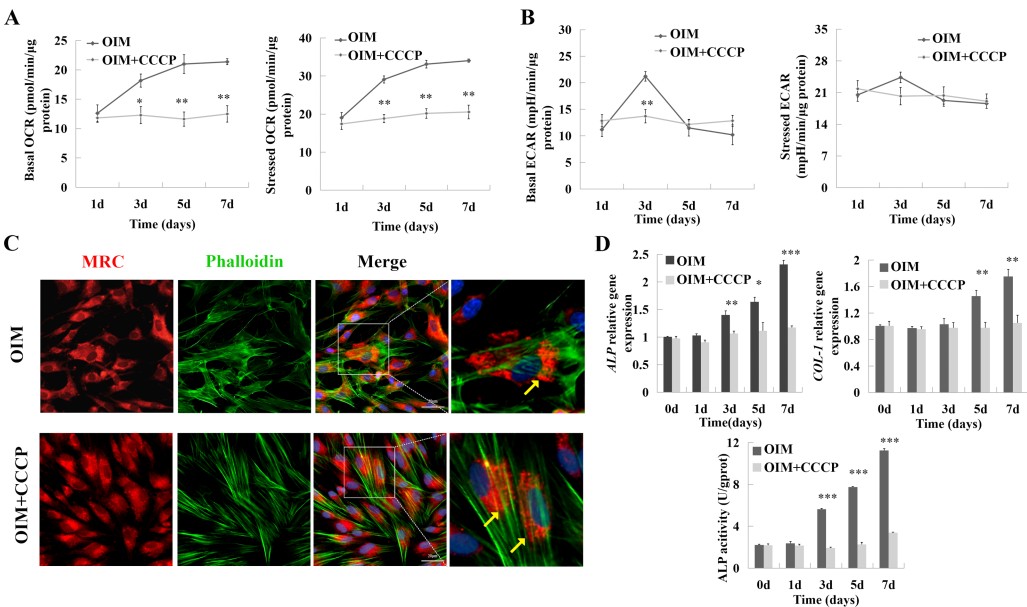

**Figure 3** **As hDPSC differentiation, cell energy phenotype shifted to mitochondrial OXPHOS.** Cells were cultured in the OIM with or without CCCP (2 μM) for 1-7 days. (A) The change in OCR was monitored over time. With the CCCP treatment, the cells' OCR was depressed compared to the control. (B) Except for day 3, the ECAR of the CCCP group had no significant change. (C) On day 3, fluorescence was used to detect the morphology of mitochondria: mitochondria - MitoTracker Red CMXRos (MRC), red; F-actin-Phalloidine, green; nucleus-Hoechst 33342, blue. The local magnification showed CCCP inhibited the form of mitochondria functional network. (D) The osteogenic differential markers, the mRNA expression of ALP and COL-1, and the activity of ALP were suppressed by CCCP. Scale bars:20 μm. Data were mean ± SD, * $p < 0.05$, ** $p < 0.01$, *** $p < 0.001$.

## The activation of AMPK impaired OXPHOS-driven differentiation

The activation of AMPK was detected in cells cultured in OIM with or without CCCP (2 μM) for 1-7 days. The ratio of AMPK phosphorylation/AMPK protein significantly increased at Days 3, 5, and 7 in the CCCP+OIM group compared to the OIM group (Fig. 4A, Day 3, 1.7 ± 0.08; Day 5, 1.55 ± 0.18; Day 7,1.79 ± 0.32). Later, whether AMPK influences the osteogenic differentiation of hDPSCs was examined. As shown in Fig. 4B, the activation of AMPK was elevated by AICAR treatment. Crucially, consistent with CCCP treatment, AICAR suppressed hDPSCs differentiation (Fig. 4C). Furthermore, the mRNA expression of *TFAM* was significantly decreased compared to the control on Days 5 and 7 after treatment with AICAR (Fig. 4D, Day 5, 3.94 ± 0.29 *vs.* 1.32 ± 0.07; Day 7, 6.08 ± 0.19 *vs.* 1.69 ± 0.4). Mitochondrial fragmentation was also observed (Fig. 4E). These results indicated that the activation of AMPK contributed to hDPSCs differential inhibition and impaired OXPHOS.

## DISCUSSION

DPSCs have been used in regenerative medicine due to their multipotent capacity to differentiate into osteoblasts, adipocytes, chondrocytes, neural cells, and even endothelial cells (*Kawashima, 2012*; *Luzuriaga et al., 2020*). Metabolism is closely related to the

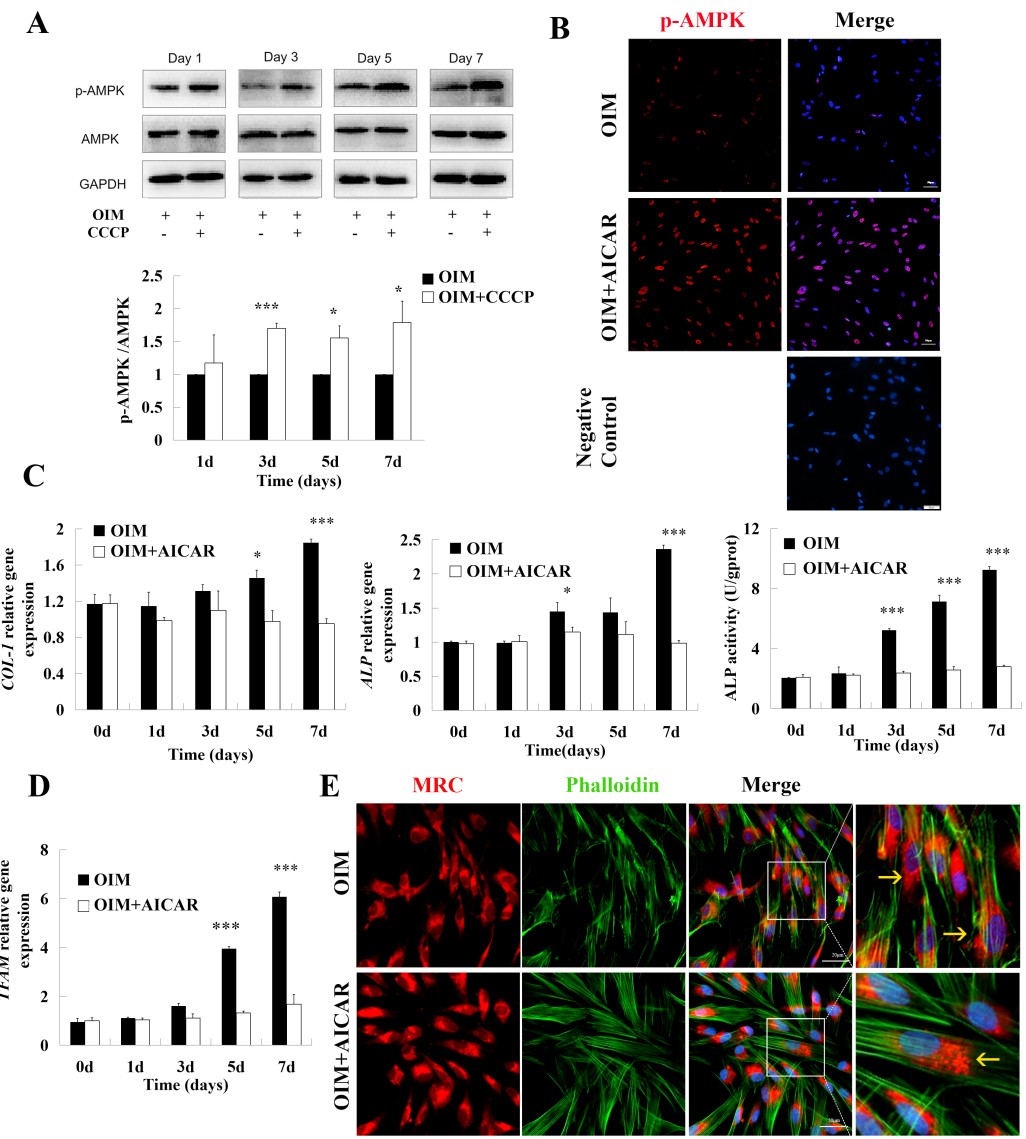

**Figure 4** **The activation of AMPK contributed to differential inhibition.** (A) Immunoblot analysis for AMPK and phosphorylated AMPK in cell lysates from hDPSCs treated with CCCP (2 μM) or OIM only for 1, 3, 5, and 7 days. GAPDH served as the standard. (B) Cells were treated with AMPK agonist AICAR (500 μM) for 3 days. The p-AMPK was observed using immunofluorescence: p-AMPK, red; nucleus, blue. Scale bar: 50 μm. (C) The ALP, COL-1 expression and ALP activity were detected to unveil the effect of AICAR on cell differentiation. (D) qPCR analysis of mitochondrial factor TFAM in AICAR-treated (500 μM) cells. (E) Mitochondrial morphology in AICAR-treated and control cells were evaluated by fluorescence staining. Mitochondria-MitoTracker Red CMXRos (MRC), red; F-actin -Phalloidine, green; nucleus - Hoechst 33342, blue. Scale bar: 20 μm. Data were mean ± SD, * $p < 0.05$, ** $p < 0.01$, *** $p < 0.001$.

physiological process of DPSCs, for example, aging and differentiation (*Macrin et al., 2019*; *Wang et al., 2016*).

This study demonstrated that a dynamic shift in glycolysis and mitochondrial OXPHOS occurred during the osteogenic differentiation of hDPSCs. Inhibiting mitochondrial

respiration with the uncoupler CCCP suppressed hDPSCs differentiation and activated the central metabolic sensor AMPK.

## Metabolic shift accompanied by hDPSCs' osteogenic differentiation

Recent studies have shown that stem cells depend primarily on anaerobic glycolysis for ATP supply while differentiating cells depend on mitochondrial aerobic respiration. The metabolic shift and mitochondrial resetting into mature bioenergetic states are considered hallmarks of stem cell differentiation (*Wanet et al., 2015*). ALP activity and gene expression of *Col-1* and *ALP* were used to evaluate the osteogenic differentiation potential of MSCs. In particular, ALP activity usually evaluates the early differentiation osteogenic potential. ALP activity and mRNA expression were upregulated during osteogenic differentiation of bone marrow mesenchymal stem cells (*Jiang et al., 2022*) and hDPSCs (*Qian et al., 2015*). In this study, the enhanced ALP activity and the increased mRNA levels of *Col-1* and *ALP* also verified the osteogenic differentiation of hDPSCs. It also verified that early osteogenic differentiation occured when the osteogenic induction lasted for 7 days.

When hDPSCs differentiated, cells presented distinct energy phenotypes. On Day 3, the level of ECAR and activities of glycolysis enzymes (HK, PK, and LDH) were elevated, showing that glycolysis increased. The results were consistent with our previous study (*Wang et al., 2016*). At this point, cells had improved glycolytic and mitochondrial function. Studies have shown that glycolysis supplies rapid energy generation and substrates for biosynthesis. Increased glycolysis can provide quick production of ATP and suitable substrates for biosynthesis, which meet the anabolic demands of initial differentiation (*Palmer et al., 2015*). This implies that differentiation requires the supply of sufficient energy and substrates. The combined effects of glycolysis and mitochondrial OXPHOS may support the tremendous energy demand on Day 3. Simultaneously, a higher level of OCR and an increased ratio of OCR/ECAR indicate a shift toward mitochondrial OXPHOS. The results showed that the enzyme activities of HK, PK, and LDH were decreased. Mitochondrial OXPHOS was the predominant source of energy production of the differentiating hDPSCs on Day 5 and Day 7 when a metabolic shift of hDPSCs was observed. The trend was similar to neural stem cells, mouse-induced pluripotent stem cells(iPSCs), and human iPSCs (*Coelho et al., 2022*; *Folmes et al., 2011*; *Prigione et al., 2010*).

Some studies have suggested that mitochondrial content, such as mitochondrial DNA (mtDNA) copy number, mitochondrial mass, and biogenesis, is closely associated with changes in cell metabolism during pathological or physical activity (*Clark & Parikh, 2020*; *Procaccio et al., 2014*). To examine the mitochondrial content-related modulating factors during hDPSCs differentiation, the mRNA expression levels of *TFAM* and *NRF1* were detected. As hDPSCs differentiated on Days 5 and 7, upregulated *TFAM* expression was observed, but *NRF1* had no significant changes. NRF-1 and TFAM are involved in proliferator-activated receptor-γ coactivator-1α (PGC-1α)/NRF-1/TFAM signaling, the main pathway controlling OXPHEN and mitochondrial biogenesis. NRF1 is a transcriptional regulator of nuclear genes that contributes to oxidative phosphorylation. TFAM prompts mtDNA transcription, replication, and maintenance (*Kang, Chu &*

*Kaufman, 2018*; *Scarpulla, 2008*). NRF-1 and TFAM are also related to the differentiation of stem cells. In a previous study, NRF-1 and TFAM were increased during the osteogenic differentiation of rat dental papilla cells (*Jiang et al., 2019*). Highly expressed TFAM in DPSCs enhanced glutamate metabolism and OXPHOS activity. Bone regeneration of DPSCs was enhanced through the activation of mitochondrial aerobic metabolism when TFAM was highly expressed (*Guo et al., 2022*). It seemed that *NRF1* was not changed during hDPSC differentiation.

## AMPK participated in differential inhibition when mitochondrial activity was attenuated

The metabolic shift is essential in regulating cellular function, particularly stem cell self-renewal, pluripotency, and plasticity (*Andre et al., 2019*). Mitochondrial biogenesis and metabolic shifts toward OXPHOS are deemed early events in multiple stem cell differentiation processes (*Khacho & Slack, 2018*; *Wanet et al., 2015*). Consistent with previous reports, our data showed that inhibiting mitochondrial function by CCCP from the beginning prevented osteogenic differentiation of hDPSCs. Mitochondrial OXPHOS was required for differentiation. CCCP-treated hDPSCs exhibited an apparent fragmentation of mitochondrial tubules, likely due to a block in mitochondrial fusion. It was reported that CCCP treatment could promote the fission of the mitochondrial network in skeletal muscle cells and a human neuroblastoma cell line (*Park, Choi & Koh, 2018*; *Seabright et al., 2020*).

The decreased intracellular ATP activated the energy sensor AMPK due to impaired OXPHOS. It was speculated that AMPK might be the direct regulator of the mitochondrial fission and fusion machinery to mediate subsequent events. As an intracellular energy sensor, the vital role of AMPK is to restore the energy balance. Once activated, AMPK induces catabolic pathways to produce energy and prevents anabolic pathways, such as lipid and protein synthesis, to save energy (*Wanet et al., 2015*). On the other hand, considerable evidence shows that AMPK regulates cell fate, including controlling stem cell pluripotency and differentiation (*Afinanisa, Cho & Seong, 2021*; *Fernandez-Veledo et al., 2013*; *Liu et al., 2021*). In this study, CCCP-treated hDPSCs exhibited an elevated level of p-AMPK, showing that the energy sensor AMPK was involved in the process of metabolism that influenced differentiation. AMPK activation was a potential modulator when the osteogenic differentiation of hDPSCs initiated.

To further verify the effect of AMPK activation, AICAR, a direct AMPK activator, was applied. AICAR resulted in mitochondrial fragmentation and differential inhibition, similar to CCCP-induced conditions. The results were consistent with previous studies in which AMPK activation has been reported to inhibit the differentiation of osteoblasts, chondrocytes, adipocytes, and myoblasts (*Bandow et al., 2015*; *Kasai et al., 2009*). For cell differentiation, numerous anabolic pathways, including the biosynthesis of proteins and lipids, are required to support the specific function of differentiated cells. Mitochondrial biogenesis and stimulation of metabolism are also needed to produce adequate energy for the processes. CCCP-induced or AICAR-induced AMPK activation may be a regulator for the cells to stop differentiation during impaired OXPHOS, which might help maintain

stemness or homeostasis. Mitochondrial dysfunction as a regulator may be a viable therapeutic target for stem cell-based therapies and interventions for cognitive defects (*Khacho, Harris & Slack, 2019*).

### Limitations

The detailed mechanism between mitochondria and osteogenic differentiation requires further investigation. Whether the metabolic shift and OPHOXS can regulate osteogenic differentiation *in vivo* needs to be further studied.

## CONCLUSION

It was demonstrated that increased mitochondrial function was indispensable for the osteogenic differentiation of hDPSCs. It was also revealed that glycolysis gradually decreased in the stage of energy supply. The mitochondrial uncoupler CCCP depressed mitochondrial OXPHOS and inhibited hDPSCs differentiation. Activation of AMPK also interferes with mitochondrial morphology, mitochondrial OXPHOS, and osteogenic differentiation. The findings helped to reveal the relationship among glycolysis, mitochondrial OXPHOS, and osteogenic differentiation. CCCP or AMPK activation may be a potential regulator to quit differentiation from impaired OXPHOS or to maintain homeostasis.

### Funding

This study was supported by the National Natural Science Foundation of China [Grant No. U20A20365 and 81970948]; and the Natural Science Foundation of Sichuan Province of China [Grant No. 2022NSFSC0613]. The funders had no role in study design, data collection and analysis, decision to publish, or preparation of the manuscript.

### Grant Disclosures

The following grant information was disclosed by the authors:
The National Natural Science Foundation of China: U20A20365, 81970948.
The Natural Science Foundation of Sichuan Province of China: 2022NSFSC0613.

### Competing Interests

The authors declare there are no competing interests.

### Author Contributions

- Lingyun Wan performed the experiments, analyzed the data, prepared figures and/or tables, authored or reviewed drafts of the article, and approved the final draft.
- Linyan Wang performed the experiments, analyzed the data, prepared figures and/or tables, authored or reviewed drafts of the article, and approved the final draft.
- Ran Cheng conceived and designed the experiments, authored or reviewed drafts of the article, and approved the final draft.
- Li Cheng conceived and designed the experiments, authored or reviewed drafts of the article, and approved the final draft.

- Tao Hu conceived and designed the experiments, authored or reviewed drafts of the article, and approved the final draft.

## Human Ethics

The following information was supplied relating to ethical approvals (i.e., approving body and any reference numbers):

Institutional Ethics Committee of West China Hospital of Stomatology granted Ethical approval to carry out the study within its facilities (WCHSIRB-D-2017-052; WCHSIRB-D-2020-075-R1).

## Data Availability

The raw measurements are available in the Supplementary Files.

## Supplemental Information

Supplemental information for this article can be found online at http://dx.doi.org/10.7717/peerj.15164#supplemental-information.

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
