# Peer review of "Metabolic shift and the effect of mitochondrial respiration on the osteogenic differentiation of dental pulp stem cells"

_PeerJ, doi:10.7717/peerj.15164_

## Round 0.1 · original submission · Major Revisions

This is an interesting study; however, based on the comments of all referees there are major and minor issues that need to be resolved. As pointed out by the reviewers, the manuscript requires revision by someone with full professional proficiency in English or an editing service to improve its clarity and readability.

Please submit a point-by-point response to the reviewers' comments in addition to your revised manuscript.

In addition to the issues pointed out by the referees, please also clarify the following:

1-Lines 247-249 indicate “significant” induction of AMPK, which seems to be through immunofluorescence (IF). Quantification method of IF needs to be stated in the Methods section.

2- In addition to the full description of stem cell confirmation requested by Reviewer #2, please provide a reference corroborating that the minimum recommended requirements for stem cell confirmation have been met in this study. This would eliminate any question regarding the origin of the isolated cells.

3- Please recheck all supplementary data to preclude non-English characters (e.g., row 27 of ampk-wb sheet in Fig 4 supplement).

4- Avoid using questions in the Results (e.g., line 247).

5- Please provide a legend for Supplementary Figure 1.

6- In addition to the comments of the reviewers, please see the annotated manuscripts provided as attachments by two of the reviewers.

Reviewer 1 ·

Basic reporting

The authors describe the metabolic shift determined by the mitochondrial respiration during the osteogenic differentiation of dental pulp stem cells. The article conform the professional standards of courtesy and expression. The manuscript maintain the professional article structure and the raw data is shared, thus the structure of the article is acceptable.
However, concerning the Literature references, the reviewer states that there is not sufficient field background/context provided. The introduction is very brief and not updated, briefly:
Line 58: The first studies of mitocondrial aerobic oxidative phosphorylation are dated from early 1957 ( doi: 10.1016/0006-3002(57)90371-2 ). Thus it should be credited instead of the "recent" reference of "Rossmann 2021".

The following sentences needs citations: Line 52: "Due to their easily isolation and great potential in tissue engineering and regenerative medicine, DPSCs are widely used in various fields"
Line 53: "Numerous studies are 54 dedicating to uncover the detailed mechanisms involved in their self-renewal ability and multi-lineage differentiation potential"

Lines 51-52, hDPSCs also has been found to be able to differentiate towards endothelial cells (doi: 10.3390/biomedicines8110483 ).

There are previous works showing the link of metabolism as early predictor of DPSCs aging (DOI: 10.1038/s41598-018-37489-4) that also shoud be included in the discussion section.
Line 88: Reference of Pantovic is not properly linked.
We encourage to the authors that the manuscript will be revised by an native English speaker to improve the use of a professional English throughout the text. For example:
Line 96: The sentence "In all, the aim of the study is to study the change of mitochondrial OXPHOS" must be reformatted.
Line 133: Please change "Enzyme activities assay" for "Enzimatic activity assay".
Line 151: Please substitute "injected" by "added".
Line 261: The sentence "Recent studies have displayed that..." change displayed for shown...

Finally there are some typo errors such lack of spaces between sentences (e.g. line 61) or final dot before references (line 63), "differention" instead of "differentiation" (line 271) that should be thoroughly reviewed.

Experimental design

The planned research fits within Aims and Scope of the journal. However in the methods section are missing all the references of the used cell culture compounds lacking the sufficient detail to be reproducible by another investigator. This is a major remark that can be solved easily.
Concerning the enzimatic activity assay, again it lacks enough description about the number of cells used, the speed of centrifugation, the volume and the total protein used for the determination.
Scale bars should be represented as micrometers (um) instead of micromolar (uM).

Validity of the findings

The data on which the conclusions are based altough provided is limitated. There is not provided a mechanistic approach in which clearly demonstrates that the metabolic shift is a prerquisite for cell differentiation.
Conclusions are well stated, linked to original research question & limited to supporting results. Indeed in the results section there is not any value of quantifiable result shown becoming only limitated to p-values.

Additional comments

The manuscript has room for improvement and we encourage the authors to provide a revised and improved version of the manuscript to be suitable for publication.
No one signs the informed consent for scientific research involving human samples. It should be at least one responsible. Is strange that editorial request an empty copy as a Confidential Supplemental Information file.

Reviewer 2 ·

Basic reporting

1) The manuscript could benefit a professional language editing service.

2) Raw data is sufficiently shared.

3) Literature references are well provided.

4)Please leave a space before using paranthesis in the entire manuscript.

5) Please use either "osteogenic differentiation medium" or "osteogenic induction medium" instead of " osteogenic mineralized medium" and make the necessary revisions in the entire manuscript.

6) There are a few terms which needs to be revised. You can see them as side notes in the annotated manuscript that I uploaded.

7) Page 4 line 18-21: “Some evidence also shows a metabolic switch from anaerobic glycolysis toward OXPHOS upon differentiation of hematopoietic stem cells (HSCs). Quiescent HSCs rely on glycolysis, while the active HSCs change to OXPHOS metabolism when cells undergo differentiation”

What is the relevance of this information on HSCs with your study? Please elaborate if there's no such study on DPSCss or other mesenchymal stem cell sourses and link this information with your study; or delete the statement.

8) Page 5 line 17-18: “AMPK activation established a metabolic barrier to impede the cellular reprogramming into iPSCs”

Please elaborate the necessity of this information or delete it.

9) Page 8 line 1-2: “Our previous results showed that hDPSCs initiated to differentiate on day 3 induced in MM. At this critical point, cells possessed increased glycolytic and mitochondrial function. “

This is more appropriate to be written in discussion. In this section, just writing that these studies were held in 3rd day would be enough.


10) Page 9 line 22: Please also mention the model of the microscope.

11) Please revise the p values in the results section by using "<" symbol and the range instead of giving the p value directly. You can give these p values in a separate table if you want.

12) Page 10, line 6-7: “3 when hDPSC started to differentiate,”

This statement is unnecessary. Indicating just the days are enough, otherwise it sounds like until the 3rd day cells stay absolutely still, and suddenly they decide to start differntiation at the 3rd day.

13) Page 14 line 4-5: “The finding was similar to other reported stem cells”
Please identicate the specie and stem cell sources of these mentioned studies.

14) Page 14 line 9-10:” the mRNA expression of TFAM and NRF1 were detected in the study”
Please briefly indicate the role and importance of these genes here. Why you're checking their expressions, what are their roles, etc.

Experimental design

1) Please mention the catalog numbers and dilutions of secondary antibodies as well.

2) Please mention the blocking serums for he necessary analyses.

3) I believe confirming that the isolated cells are indeed stem cells, and confirming the osteogenic differentiation are important to show. Please briefly write these methods and their results in the original manuscript. Images could stay as supplementary files if needed, yet I believe confirming these studies are important to show.

4)Please use a uniform style when writing mililiters. Please use either ml or mL.

5) Page 7 line 16-17: “In some experiments, cells were treated with 5-aminoimidazole-4-carboxamideribonucleotide”

In which experiments? Please indicate.

6) For real time PCR analysis:
Which RNA quantification method was used?
Please also indicate how did you calculate the delta ct and fold change in expression values.
Please indicate the passage of hDPSCs used for PCR studies.

7) Page 8 line 22, Page 9 line 1-9: Please prepare a table for primers instead of writing them like this and please also add the Tm values for each gene as well.

8) In fluoroscence microscopy: please mention if the cells were seeded in 4 well plates, 6 well plates, or any other?

9) Page 9 line 19 (and in WB also): Did you use only BSA for blocking? Since I cannot see the catalog number of secondary antibodies I’m not sure if it’s appropriate to make a blocking with BSA.

10) For p-AMPK antibody (IF), it says that this antibody is only tested for WB in the manufacturer’s website. Please provide a citation for its use in IF or provide a positive and negative control stainings for this antibody.

Validity of the findings

1) Please mention if the increases/decreases are statistically significant or not in the results section.

2) Page 10 line 1-2: “To further understand the hDPSC differentiation, ALP activity was also monitored”

Do authors performed ALP eyzme assay only to further understand the differentiation or to evaluate the ALP production levels during differentiaon?


5) Page 14 line 11-13: “As hDPSCs differentiation went onto day 5 and 7, up-regulated TFAM expression were observed, which might induced due to increased mtDNA content and increased mitochondrial biogenesis”
What does this increased TFAM expessions telling us? Why is this important?

6) Page 14 line 14-15:” The reason might be related to the difference of cell types.”
Please elaborate this statement. Are there any other stem cell types that NRF1 was found to be involved with this process? Why do you think this might be due to the difference of cell types?

7) Page 15 line 3-4: “likely due to a block in mitochondrial fusion. “
Please elaborate this statement too. What made you think this way?

8) There were no discussion related to Col-1 and ALP gene expressions or the enzyme acitivity levels of ALP, HK, PK and LDH. Were the results of these studies irrelevant with your study? if so please discard everything about them or please use them in your discussion to state their importance and how the results of these studies served you to reach this conclusions.

9) In the manuscript authors made statements about the role of mithocondrial function in cell differentiation; yet the study is based only on osteogenic differentiation. Therefore, all the satements of "cell differentiation" should be changed as "osteogenic differentiation".

Additional comments

For my detailed comments please see the uploaded PDF file.

Annotated reviews are not available for download in order to protect the identity of reviewers who chose to remain anonymous.

·

Basic reporting

The authors were aimed to investigate the change of mitochondrial OXPHOS and glycolysis during hDPSCs differentiation and its possible regulating factors. Although only hDPSCs were focused in this study, the comparative analyses might be very exciting.

The recommendations with the study were given below:
-It is recommended to write in other p values used in the figures (line 198).
-Reference format should be corrected (line 89)
-Figure 4-E X column text should be made clearly visible
-Additionally; some minor spelling mistakes marked in purple on PDF.

Experimental design

The experimental approach was well designed and the results were clearly represented.

Validity of the findings

Conclusions are well stated, linked to original research question & limited to supporting results.

---

## Round 0.2 · Minor Revisions

Thank you for applying the requested changes and revising the manuscript, which has improved significantly. As pointed out by one of the reviewers, there are still some issues that require attention, before the paper can be considered for publication. Please submit a point-by-point response to the reviewer’s comments in addition to your revised manuscript.

Reviewer 1 ·

Basic reporting

The authors have strongly improved this version of the manuscript. Now the introduction address the correct references to credit the discoveries of the initial authors and there is a balance of recent reviews that cover sufficient field background.

Experimental design

Methods section has strongly improved providing the requested references of the compounds as suggested by the reviewer.

Validity of the findings

With the addition of the information, data values and statistics suggested by the reviewers, now the manuscript has strongly improved fulfilling the requirements of this section.

Additional comments

The manuscript has strongly improved to be suitable for publication.

Reviewer 2 ·

Basic reporting

1) Authors had mentioned their previous research in 2 parts in the manuscript (Line 155, Lines 291-292) but forgot to cite that work. Citations should be added for their previous study in these parts.

2) There are a few typos and punctuation errors.

3) Line 308-320: Authors had cited a few previous articles describing the role of NRF-1 and TFAM in stem cell differentiation. These studies are referring to neurogenic and cardiomyogenic differentiation of stem cells. Authors are sugessted to use articles on osteogenic differentiation rather than these differentiations since the focus of this manuscript is on osteogenic differentiation.

Experimental design

1) Authors were asked a few questions about the PCR. Thank you for your responses in this part. Yet, authors forgot to mention the RNA quantification method. Authors are suggested to share the RNA quantification method (nanodrop, Qubit, etc.?).

2) Shouldn't authors had performed a blocking with goat serum for IF? For WB use of BSA or non-fat milk is a standard application, yet in IF authors should perform a blocking regarding with the host specie of the secondary antibody.

Validity of the findings

1) For anti p-AMPK IF staining, authors were asked for either a positive control or a citation mentioning the use of this primary antibody since the manufacturer listed only WB as the tested application. Authors had presented a negative negative control staining (with a merged image). In order to see if this antibody works in IF applications and specific to the targeted protein, we should see a positive control staining. Negative control proves that your secondary antibody is not binding a non-specific target. To confirm primary antibody, we should either see a citation regarding the previous use of this antibody or a positive control. The positive control samples are listed by the manufacturer is listed below:
HEK-293 and MOLT-4 whole cell lysates; Mouse heart lysate; Rat brain lysate.
However, finding just a previous example of IF application with this antibody and citing it would be enough to confirm. The "citeab" website might be useful to find antibody citations.

2) In my previous review I asked "Do authors performed ALP eyzyme assay only to further understand the differentiation or to evaluate the ALP production levels during differentiaon? " This question was asked because authors had stated that they performed this assay "to further understand the differentiation" in their manuscript. I asked that because ALP enzyme activity is important for determining the early osteogenic activity of cells during especially the early osteogenic differentiation process. This analyses should be used for comparing the early osteogenic response of the cells. Authors are recommended to use the articles they had cited in their rebuttal and make a little comparison about the early osteogenic response of the cells in their experimental groups and use the results of that comparison for a little discussion. Otherwise the enzyme activity assays are not giving any special information to use if they're not discussed in a way they should be.

Additional comments

Thank you for your efforts and careful responses to the previous inquiries. I only have a few minor questions which you can see above. After adding a little discussion about the enzyme activity assays and providing a citation or a positive control for anti-p-AMPK IF staining,

---

## Round 0.3 · accepted · Accept

I am pleased to accept your article.

Reviewer 2 ·

Basic reporting

Authors responded all previous inquiries in a positive manner and performed the necessary revisions.

Experimental design

Authors responded all previous inquiries in a positive manner and performed the necessary revisions.

Validity of the findings

Authors responded all previous inquiries in a positive manner and performed the necessary revisions.

Additional comments

Authors responded all previous inquiries in a positive manner and performed the necessary revisions.